# JAK/STAT Inhibition Normalizes Lipid Composition in 3D Human Epidermal Equivalents Challenged with Th2 Cytokines

**DOI:** 10.3390/cells13090760

**Published:** 2024-04-29

**Authors:** Enrica Flori, Alessia Cavallo, Sarah Mosca, Daniela Kovacs, Carlo Cota, Marco Zaccarini, Anna Di Nardo, Grazia Bottillo, Miriam Maiellaro, Emanuela Camera, Giorgia Cardinali

**Affiliations:** 1Laboratory of Cutaneous Physiopathology and Integrated Center of Metabolomics Research, San Gallicano Dermatological Institute, IRCCS, 00144 Rome, Italy; enrica.flori@ifo.it (E.F.); alessia.cavallo@ifo.it (A.C.); sarah.mosca@ifo.it (S.M.); daniela.kovacs@ifo.it (D.K.); anna.dinardo@ifo.it (A.D.N.); grazia.bottillo@ifo.it (G.B.); miriam.maiellaro@ifo.it (M.M.); giorgia.cardinali@ifo.it (G.C.); 2Genetic Research, Molecular Biology and Dermatopathology Unit, San Gallicano Dermatological Institute, IRCCS, 00144 Rome, Italy; carlo.cota@ifo.it (C.C.); marco.zaccarini@ifo.it (M.Z.)

**Keywords:** 3D skin model, Th2 cytokines, skin lipidomics, JAK/STAT, atopic dermatitis

## Abstract

Derangement of the epidermal barrier lipids and dysregulated immune responses are key pathogenic features of atopic dermatitis (AD). The Th2-type cytokines interleukin IL-4 and IL-13 play a prominent role in AD by activating the Janus Kinase/Signal Transduction and Activator of Transcription (JAK/STAT) intracellular signaling axis. This study aimed to investigate the role of JAK/STAT in the lipid perturbations induced by Th2 signaling in 3D epidermal equivalents. Tofacitinib, a low-molecular-mass JAK inhibitor, was used to screen for JAK/STAT-mediated deregulation of lipid metabolism. Th2 cytokines decreased the expression of *elongases 1*, *3*, and *4* and *serine-palmitoyl-transferase* and increased that of *sphingolipid delta(4)-desaturase* and *carbonic anhydrase 2*. Th2 cytokines inhibited the synthesis of palmitoleic acid and caused depletion of triglycerides, in association with altered phosphatidylcholine profiles and fatty acid (FA) metabolism. Overall, the ceramide profiles were minimally affected. Except for most sphingolipids and very-long-chain FAs, the effects of Th2 on lipid pathways were reversed by co-treatment with tofacitinib. An increase in the mRNA levels of *CPT1A* and *ACAT1*, reduced by tofacitinib, suggests that Th2 cytokines promote FA beta-oxidation. In conclusion, pharmacological inhibition of JAK/STAT activation prevents the lipid disruption caused by the halted homeostasis of FA metabolism.

## 1. Introduction

Th2 cytokines drive inflammation in atopic dermatitis (AD), which is a multifactorial disorder characterized by dysfunction of the epidermal permeability barrier (EPB) in the stratum corneum (SC). In AD, the impairment of the EPB is caused by the disruption of epidermal proteins and lipids, both produced by keratinocytes [1]. It has long been debated whether the impairment of the skin barrier is a direct consequence of the Th2 cytokine environment [2], genetic predisposition, altered microbiome, or environmental insults [3]. It is commonly acknowledged that the proteins essential for proper skin differentiation, such as filaggrin (FLG), loricrin (LOR), and involucrin (IVL), are suppressed by the IL-4 and IL-13 cytokines. Notably, proteins of the EPB present decreased levels in acute AD skin lesions [4,5,6,7,8]. However, more research is needed to completely understand the mechanisms underlying the lipid alterations observed in AD.

We propose that Th2 cytokines cause impairment of the EPB by interfering with the mechanism that leads to the synthesis of barrier lipids. By employing 3D human epidermal equivalents, we sought to investigate the possible involvement of the JAK/STAT axis in the alteration of lipid metabolism in the skin barrier. The key players involved in the signaling pathways from Th2 to AD development are summarized in Table 1, which was modified from Amano et al. [9]. When Th2 cytokines bind to the IL-4Rα/IL-13Rα1 receptor complex on keratinocytes, they initiate the JAK/STAT signaling cascade. Seven transcription factors make up the STAT family, each phosphorylated and activated by different JAKs [10,11,12]. After dimerization, STATs translocate from the cytosol into the nucleus, where they promote gene transcription. The JAK/STAT pathway is essential to the exaggerated Th2 cell response observed in AD [11]. JAK1- and JAK3-associated receptors modulate the IL-4 signaling pathway via phosphorylation of STAT3, STAT5, and STAT6 [10]. In turn, Th2 cell differentiation is enhanced by IL-4, which stimulates the release of additional cytokines [11]. Recent data point to the role of the JAK/STAT pathway in Th2-cytokine-mediated modifications of the lipid profile of the EPB [13]. Several studies have demonstrated the benefits of inhibiting the JAK/STAT pathway in AD [14,15,16]. Most JAK inhibitors screened for AD treatment affect the IL-4 pathway [17].

In the first generation of small molecules designed to block JAKs selectively, tofacitinib has been indicated for topical treatment for mild to moderate AD. Currently, several JAK inhibitors are candidates for approval by the US Food and Drug Administration (FDA) for application in AD management [18]. Tofacitinib modulates cytokine signals critical in the progression of immune and inflammatory processes and affects the innate and adaptive immune responses. Tofacitinib is an inhibitor of JAK1 and JAK3 and, to a lesser extent, of JAK2 [19]. Improvement in Eczema Area and Severity Index (EASI) scores has been observed after four weeks of topical treatment with tofacitinib [20]. Studies in vitro and in vivo show improved keratinocyte differentiation and EPB function [12]. Tofacitinib has been proven to modulate the activity of the JAK/STAT pathway in keratinocytes [21]. Our study aimed to investigate the role of the JAK/STAT axis in the deregulation of lipid homeostasis induced by Th2 cytokines. Tofacitinib, which was commercially available as a chemical at the time of the study, was used to inhibit JAK/STAT pharmacologically.

## 2. Materials and Methods

### 2.1. Materials

The immortalized human keratinocyte cell line Ker-CT (ATCC^®^ CRL-4048TM) was acquired from ATCC (Manassas, VA, USA). M154, calcium chloride (0.2 M), human keratinocyte growth supplements (HKGS), L-glutamine (2 mM), penicillin (100 U/mL), streptomycin (100 µg/mL), fetal bovine serum (FBS), trypsin/EDTA, and D-PBS were purchased from Invitrogen Technologies (Monza, Italy). An Aurum^TM^ Total RNA Mini kit, SYBR Green PCR Master Mix, and Bradford reagent were obtained from Bio-Rad (Milan, Italy). A RevertAid^TM^ First Strand cDNA synthesis kit was obtained from Thermo Fisher Scientific (Monza, Italy). IL-4 and IL-13 were obtained from Peprotech (Cranbury, NJ, USA). GAPDH antibody (G9545) was from Sigma-Aldrich (Milan, Italy). Antibodies against STAT1 (#14994), phospho-STAT1 (Tyr701) (#9167), STAT3 (#9139), phospho-STAT3 (Tyr105) (#9145), STAT6 (#9362), and phospho-STAT6 (Tyr641) (#9361) as well as a secondary anti-mouse IgG HRP-conjugated antibody and an anti-rabbit IgG HRP-conjugated antibody were purchased from Cell Signaling (Danvers, MA, USA). Anti-IVL (ab53112), anti-FLG (ab24584), anti-cytokeratin 10 (ab76318), and anti-LOR (ab85679) antibodies were purchased from Abcam (Cambridge, UK). Anti-ELOVL1 (NBP312302) was purchased from Novus Biologicals™ (Centennial, CO, USA). Amersham ECL Western Blotting Detection Reagent was obtained from GE Healthcare (Buckinghamshire, UK). A protease inhibitor cocktail was from Roche (Mannheim, Germany). RIPA lysis buffer and cocktails of broad-spectrum protease and phosphatase inhibitors were purchased from Boster Biological Technology Co. (Pleasanton, CA, USA).

### 2.2. Chemicals

Deuterium-labeled ceramide LIPIDOMIX^®^ Mass Spec Standard Solution, EquiSPLASH™ LIPIDOMIX^®^ Mass Spec Standard Solution, and N-palmitoyl-d31-D-erythro-sphingosine (d31-Cer16:0, MW 569.1) were purchased from Avanti Polar Lipids (Alabaster, AL, USA). Deuterated cholesterol-2,2,3,4,4,6-d6 (d6-CH MW 392), deuterated cholesterol sulfate sodium salt (d7-CHS, MW 495), and hexadecanoic-9,9,10,10,11,11,12,12,13,13,14,14,15,15,16,16,16-d17 acid (d17-PA, MW 273), glyceryl trihexadecanoate-d98 (d98TG 48:0, MW 906), and n-hexadecyl-1,1,2,2-d4 hexadecanoate-16,16,16-d3 (d7WE, MW 488) were purchased from CDN Isotopes Inc. (Pointe-Claire, Quebec, Canada). Details on the internal standards used are reported in Appendix A. HPLCMS-grade ethyl acetate, acetone, and chloroform were purchased from Carlo Erba (Milan, Italy). HPLCMS-grade acetonitrile, isopropanol, and methanol were purchased from Biosolve (Chimie SARL, Dieuze, France; BV, Valkenswaard, Netherlands). Ultra-HPLCMS-grade water was purchased from LiChrosolv by Merck (Darmstadt, Germany). Butylated hydroxytoluene (BHT) and the mobile-phase modifiers ammonium formate (NH_4_COOH) and ammonium fluoride (NH_4_F) were purchased from Sigma Aldrich (Milan, Italy). The mass calibration solution was prepared from Agilent Technologies Tuning mix (HP0321 solution, Agilent Technologies, Santa Clara, CA, USA) upon dilution in acetonitrile.

### 2.3. Culture of 3D Epidermal Equivalents and Stimulation with Cytokines and a JAK1/3 Inhibitor

The Ker-CT cell line, consisting of immortalized human keratinocytes, was maintained at 37 °C under 5% CO_2_ in the defined medium M154 with HKGS, 2 mM L-glutamine, 100 U/mL penicillin, 100 µg/mL streptomycin, and 100 µM CaCl_2_. For routine cell culture, cells were passaged when 60–70% confluence was reached. The Ker-CT cell line was used to generate 3D human epidermal equivalents (HEEs). Briefly, Ker-CT cells were seeded on cell culture inserts (Thermo Scientific, Roskilde, Denmark; 0.4 µm pore size; 2 × 10^5^ cells per insert), maintained submerged for 3 days in CnT-Prime Epithelial Culture Medium (CnT-PR) (CellnTEC, Bern, Switzerland), and then switched to CnT-Prime 3D Barrier Medium (CnT-PR-3D) in an air–liquid condition for 12 days. Fresh medium was replaced every other day. To reproduce in vitro the effects of Th2 cytokines as observed in AD skin, IL-4 (10 ng/mL) and IL-13 (10 ng/mL) were added during the last 5 days of air–liquid culture. HEEs were treated with Th2 cytokines with or without 2 µM tofacitinib diluted in the medium 1 h before the addition of cytokines. HEE samples were processed for gene and protein expression analyses, lipidomic profiles, and immunohistochemistry. For routine histological procedures, samples were formalin-fixed and paraffin-embedded for hematoxylin and eosin (H&E) staining, morphometry and immunofluorescence analyses.

### 2.4. Isolation of RNA and Analysis of mRNA by Real-Time RT-PCR

Total RNA was extracted from 3D HEEs using the Aurum^TM^ Total RNA Mini kit and stored at −80 °C until use. Following DNAse I treatment, cDNA was synthesized using a mix of oligo-dT and random primers and a RevertAidTM First Strand cDNA synthesis kit. Real-time RT-PCR was performed in 10 μL sample volume with SYBR Green PCR Master Mix and 200 nM concentration of each primer. The sequences of the primers used are reported in Appendix A. Reactions were carried out in triplicate using a CFX96 Real Time System (Bio-Rad Laboratories Srl). Melting curve analysis was performed to confirm the specificity of the amplified products. The relative expression of mRNA was normalized to the expression of GAPDH mRNA by the change in the Δ cycle threshold (ΔCt) method and calculated based on 2^−ΔCt^. The cycle time (Ct) read of GAPDH mRNA confirmed that the expression level of the gene was stable in all treatment groups.

Appendix A reports the mean values and standard deviations (SDs) of the fold change (FC) in mRNA expression of inflammatory and lipid genes in treated HEEs compared to vehicle controls. Results were expressed as the FC between treatment and vehicle (taken as 1-fold). Data are represented as the mean ± SD of three independent experiments.

### 2.5. Western Blot Analysis

HEEs were lysed in RIPA buffer supplemented with a protease/phosphatase inhibitor cocktail and then sonicated. Total lysates were centrifuged at 12,000 rpm for 10 min at 4 °C and then stored at −80 °C until analysis. Following spectrophotometric protein measurement, equal amounts of protein were resolved on acrylamide SDS-PAGE and transferred onto a nitrocellulose membrane (Amersham Biosciences, Milan, Italy). Protein transfer efficiency was checked with Ponceau S staining (Sigma-Aldrich). Membranes were first washed with water, blocked with EveryBlot Blocking Buffer (Bio-Rad Laboratories Srl, Milan, Italy) for 10 min at room temperature and then treated overnight with primary antibodies at 4 °C, according to instructions. Secondary anti-mouse or anti-rabbit IgG HRP-conjugated antibodies were used. Antibody complexes were visualized using enhanced chemiluminescence (ECL). A subsequent hybridization with anti-GAPDH was used as the loading control. Protein levels were quantified by measuring the optical densities of specific bands using UVITEC Imaging System (Cambridge, UK). Results were expressed as the FC relative to vehicle (taken as 1-fold). Data represented the mean ± SD of three independent experiments.

### 2.6. Histology, Morphometry, and Immunofluorescence Analysis of HEEs

Histological and morphometric analyses were performed on de-paraffinized HEEs sections stained with hematoxylin and eosin (H&E). Sections were analyzed serially by recording stained images with a cooled CCD color digital camera (Zeiss, Oberkochen, Germany). The blue edition of the Zen 2.6 software (Zeiss) was used for the evaluation of epidermal and SC thickness. At least 100 measurements were taken on the images acquired under the different experimental conditions for either the epidermis or the SC. Sample sections were analyzed along their entire length. The results were expressed as the average thickness value ± SD obtained from three different experiments. For the immunofluorescence analysis, sections were dewaxed in xylene and rehydrated with graded ethanol in PBS. The antigen retrieval was obtained by heating the sections at 95.7 °C in slightly acidic conditions (pH 6). Then, the sections were blocked for 15 min with 5% normal goat serum in PBS and incubated overnight at 4 °C with the following primary antibodies: anti-IVL (1:200 in PBS), anti-LOR (1:300 in PBS), and anti-FLG (1:200 in PBS). The primary antibodies were visualized by incubating the sections for 2 h at room temperature with the following secondary antibodies: anti-rabbit IgG-Alexa Fluor 555 conjugated antibody (1:800 in PBS) and anti-mouse IgG-Alexa Fluor 488 conjugated antibody (1:800 in PBS) (Cell Signaling). Sections were mounted using ProLong mounting with DAPI (TermoFisher). For the immunohistochemical staining of ELOVL1, the tissue sections were dewaxed, processed for antigen-retrieval by heating at 95.7 °C in alkaline solution (pH 9), and then incubated with the primary antibody (1:200 in PBS). The staining was visualized by the Thermo Ultravision Quanto Detection System HRP, using 3,3′-diaminobenzidine as substrate chromogen. All the sections were counterstained with hematoxylin. Images of stained sections were recorded using a CCD camera on a Zeiss microscope (Axioskop 2 Plus), and the signal intensity was quantified using Zeiss Zen 2.6 (blue edition) software for image analysis.

### 2.7. Lipid Extraction

The extraction of lipids was slightly adapted from our previous study [22]. The epidermal sheet was isolated from the insert following incubation overnight with dispase. HEEs were dried on absorbent paper, then, the epidermal sheet was extracted with water/methanol/chloroform (1/3.32/1.66 *v*/*v*/*v*) in presence of a mixture of deuterium labelled internal standards with the individual concentration described in the Appendix A. The internal standards included deuterated cholesterol, fatty acids (FAs), ceramides, triglycerides (TGs), and phospholipids (the latter ones from the EquiSPLASH-LIPIDOMIX) in 20 µL of a methanol solution containing BHT 1.2 mM to prevent autoxidation. The lipid extract was dried under nitrogen flow and suspended in 200 µL chloroform/methanol 2/1 *v*/*v* prior to analysis.

### 2.8. GCMS Analysis

GCMS served for the semi-quantitative measurement of free FAs (FFAs), fatty alcohols (FOHs), and cholesterol following derivatization. Specifically, the analyzed saturated FFAs (SFAs) were 12–26 carbon atoms long. Four branched SFAs (iso- and anteiso-branched FAs) with carbon numbers between 15 and 17 were also detected. Ten monounsaturated FAs (MUFAs) with chain lengths between C14 and C24 and the polyunsaturated FA (PUFA) linoleic acid (FA 18:2) were detected. Twenty microliters of the lipid extract obtained from the 3D epidermis models were dried under nitrogen and derivatized with 40 µL of BSTFA-1% trimethylchlorosilane mixture in pyridine. The reaction was carried out at 60 °C for 60 min to produce the trimethylsilyl (TMS) derivatives of most lipids. The GCMS analysis was performed with the 8890 GC system combined with the 5977B Series MSD single quadrupole (Agilent Technologies, Santa Clara, CA, USA). Helium was used as the carrier gas at the flow rate 1.2 mL/min. The analysis was conducted on the HP-5MS UI fused silica column (30 m × 0.250 mm internal diameter × 0.25 µm film thickness, chemically bound with a 5%-phenyl-methylpolysiloxane phase (Agilent Technologies, Santa Clara, CA, USA). The GC oven program was as follows: initial temperature 80 °C, hold for 2 min, 280 °C at 33 min, 310 °C to final run time of 49 min. Samples were analyzed in scan mode following EI ionization [23].

### 2.9. LC Separations 

Reversed-Phase High-Performance LC (RP-HPLC) was applied to the separation of relatively hydrophobic lipids [24]. TGs and diglycerides (DGs) were detected under positive electrospray ionization (+ESI) conditions; ceramides, cholesterol sulfate, and long chain SFAs (27–30 carbon atoms) were analyzed under negative ESI (−ESI) conditions. Hydrophilic Interaction Liquid Chromatography (HILIC) was used to separate polar and hydrophilic lipids. Quantification of phosphatidylcholines (PCs), lysophosphatidylcholines (LPCs), and sphingomyelins (SMs) was performed in +ESI mode. Phosphatidylethanolamines (PEs), ether-linked phosphatidyl-ethanolamine (PE O-), phosphatidylinositols (PIs), and phosphatidylglycerols (PGs) were detected in −ESI mode. RPLC separation was conducted on the Infinity II 1260 series HPLC system equipped with a degasser, a quaternary pump, an autosampler, and a column compartment (Agilent Technologies, Santa Clara, CA, USA). The RP-HPLC separation was performed using the Zorbax Eclipse Plus C18 column (2.1 × 50 mm, 1.8 µm particle size) (Agilent Technologies, Santa Clara, CA, USA). The maximum operating pressure was 600 bar/9000 psi. Cell extracts were eluted with a binary gradient of (A) 0.2 mM NH_4_F in water (18.2 Ω) and (B) 0.2 mM NH_4_F in methanol/isopropanol 80/20 [25]. Following a hold time of 2 min in 40% B, the gradient 40–99% occurred between 2.0 and 36.0 min; 99% B was held from 36.0 to 46.0 min. The mobile phase returned to 40% B between 46.0 and 48.0 min. A 10 min post-run time of 40% B was included. The column was maintained at 60 °C with a thermostat; the flow rate was 0.3 mL/min; and the injection volume was 0.6 µL and 1 µL in +ESI and −ESI modes, respectively. HILIC separation was performed with a HALO HILIC column, 2.1 × 50 mm, 2.7 µm particle size, with maximum operating pressure at 600 bar/9000 psi (Advanced Materials Technology, Phoenix, AZ, USA). The column temperature was set at 40 °C. The mobile phase consisted of (A) aqueous solutions of 5 mM NH_4_COOH in water (18.2 Ω) and (C) acetonitrile. The elution program was 98% C, 0–1.0 min; 98–80% C, 1.0–18.0 min; 80% C, 18.0–20.0 min; 80–98% C, 20.0–21.0 min, 98% C, 21.0–22.0 min. A 10 min post run of the initial condition was added. The mobile phase flow rate was 0.4 mL/min, the injection volume was 0.4 µL and 1 µL in +ESI and −ESI modes, respectively.

### 2.10. HRMS

A Quadrupole Time-of-Flight (QTOF) mass spectrometer 6545 was interfaced to the the HPLC instrument using the ESI Dual Agilent Jet Stream (AJS) interface (Agilent Technologies, Santa Clara, CA, USA). Nebulization and desolvation were supported by gaseous nitrogen. The ion source gas temperature was set at 200 °C and the flow rate at 12 L/min; the nebulizer pressure was 40 psi. Sheath gas temperature was set at 350 °C; sheath gas flow rate was 12 L/min. The capillary voltage parameter was 4000. The fragmentor voltage was 120 V, and the skimmer voltage was 40 V. In the preliminary screening, data independent acquisition (DIA) was performed in all-ions MS/MS mode, at three collision energy (CE) values, i.e., 0, 20, and 40 eV. For the structural elucidation, data dependent acquisition (DDA) was accomplished with targeted MS/MS. The *m*/*z* range for MS and MS/MS was 59–1700 at a mass resolving power of 40.000. Internal mass calibration for accurate mass measurement used *m*/*z* 121.0509 and *m*/*z* 922.0098 in +ESI; reference ions in –ESI were *m*/*z* 112.9856 and *m*/*z* 966.0007 using NH_4_COOH; *m*/*z* 119.0363 and *m*/*z* 940.0015 using NH_4_F.

### 2.11. Data Processing and Statistical Analysis

Data derived from Western blot analysis, real-time RT-PCR, histology, morphometry, and immunofluorescence analysis were represented as the mean ± SD of three independent experiments. The values were expressed as relative to the vehicle (taken as 1). Statistical significance was assessed using paired Student’s *t*-test or ANOVA followed by Tukey’s multiple comparisons test using GraphPad Prism (GraphPad Software version 10.1.2). The threshold level of significance was *p* < 0.05. GCMS data were acquired using Agilent MassHunter GC/MSD 5977B acquisition software (version 10.0, BUILD 10.0.384.1) and processed with Agilent MassHunter Workstation Software Quantitative Analysis (version 10.1). LC-HRMS data were acquired and deconvoluted using the MassHunter Data Acquisition Software (B.09.00, Agilent Technologies). The data acquired by LCMS were processed with Agilent MassHunter Workstation Profinder (version 10.0). In keeping with the current recommendation in lipidomics practice [26,27], all data were obtained by comparing each lipid’s response to that of the corresponding labelled internal standard (e.g., FFAs vs. d17-PA, cholesterol vs. d6-cholesterol, etc.) and multiplying by the deuterated internal standard pmole. In turn, mole amounts were normalized by the protein content and reported as pmol/mg protein.

The multivariate statistical analysis was performed in Agilent MassHunter Mass Profiler Professional (MPP) (version 15.1). The mole amounts calculated for the lipid species were organized in a template (.csv) and imported into the software to determine relevant differences of compounds across groups. The data on a linear scale were log2 transformed, and a baseline to the same-experiment controls was applied. To create interpretations, samples were divided into groups according to the specific treatment, i.e., vehichle, tofacitinib, Th2 cytokines, and their combination. One-way analysis of variance (one-way ANOVA) was performed. The FC ratio was calculated in selected conditions. Tukey HSD was used as the post hoc test to explore differences between treatments. For the multiple testing correction Storey’s approach was chosen using the bootstrap method with a q-value cut-off ≤ 0.05. Data were further analyzed using XLSTAT 2020.1.2 (Addinsoft, New York, NY, USA). Continuous variables were represented as mean ± SD. Significant differences between and within multiple groups were examined using Kruskal–Wallis. The Dunn method was used for multiple pairwise comparisons and Bonferroni’s correction of the significance level was applied. *p*-Values were calculated using the approximation of the distribution of K by a chi-square distribution with (k−1) degrees of freedom. Differences and correlations were considered statistically significant with *p* ≤ 0.05.

## 3. Results

### 3.1. Effects of JAK/STAT Inhibition by Tofacitinib on Th2 Cytokine-Mediated Changes in Epidermal Morphology and Barrier Protein Expression

To reproduce the effects of Th2 cytokines in vitro, human epidermal equivalents (HEEs) were treated with IL-4 and IL-13, each at a concentration of 10 ng/mL, for five days. Conventional H&E staining of the 3D HEEs treated with the cytokines revealed histological features resembling AD epidermis in vivo. Occurrence of spongiosis was demonstrated by the expanded space between adjacent keratinocytes (Figure 1A, arrows). Morphometric analysis and quantitative evaluation of the thickness of both viable epidermis and SC showed a slight but statistically significant increase in the thickness of epidermal portion in Th2-treated HEEs compared to control. Tofacitinib partially counteracted the increased epidermal thickness induced by Th2 cytokines (Figure 1B). No significant modulation of the mRNA expression of the early differentiation marker *K10* and the late differentiation markers *LOR* and *IVL* was detected after stimulation with Th2 cytokines (Figure 1C). According to previous evidence [9,28], Th2 cytokines caused a significant decrease in the mRNA levels of *FLG* and *CASP14*, which is the enzyme crucial for the *FLG* degradation process [29]. Tofacitinib treatment abolished the reduction of *FLG* and *CASP14* mRNA observed in Th2-treated HEEs (Figure 1C). Consistent with the mRNA levels, protein quantification by both Western blot (Figure 1D) and immunofluorescence analyses (Figure 1E,F) showed a significant decrease in FLG expression in Th2-treated HEEs, which was prevented by tofacitinib. A similar result was obtained when profilaggrin (pro-FLG) and FLG expression were analyzed separately in the stratum granulosum (SG) and the stratum corneum (SC) (Figure 1G).

### 3.2. JAK/STAT Inhibition Counteracts the Induction of Inflammatory Genes in Th2-Treated Epidermal Equivalents

To evaluate the effect of tofacitinib on STAT signaling, Western blot analysis of native STAT1, STAT3, and STAT6 and their phosphorylated forms was performed (Appendix A). As expected, the stimulation with Th2 cytokines upregulated the phosphorylation of STAT3 and STAT6; tofacitinib efficiently counteracted their activation. No significant modulation of STAT1 phosphorylation was observed after Th2 exposure (Appendix A). The expression of genes involved in the inflammatory response was evaluated in HEEs and Th2-treated HEEs in the presence of tofacitinib. The compound hindered the Th2-mediated increase in the mRNA expression of the IL-1α, IL-1β, IL-6, and IL-8 cytokines; the CCL26 chemokine; and podoplanin (PDPN), a keratinocyte glycoprotein upregulated via JAK/STAT signaling (Figure 2A) [30]. 

### 3.3. Modulation of Lipid Genes by Th2 Cytokines and Counteracting Effects of Tofacitinib

Next, we investigated the activity of IL-4/IL13 in the synthesis of sphingolipids. Th2 cytokines modulated several genes involved in lipid metabolism (e.g., elongation of FFAs and synthesis of sphingolipids), as determined by real-time RT-PCR. A significant reduction in the mRNA levels of FA elongases (ELOVLs) 1, 3, and 4 was observed in Th2-treated HEEs (Figure 2B). Co-treatment with tofacitinib prevented the reduction of ELOVL3 and ELOVL4 mRNA induced by Th2 cytokines. The decrease of ELOVL1 expression following Th2 cytokines and the recovery in the presence of tofacitinib were also observed at the protein level (Figure 2C–E). The expression of genes involved in the biosynthesis of sphingolipids, such as serine palmitoyltransferase (SPT), and sphingolipid delta(4)-desaturase (DEGS2), showed an opposite response to Th2 treatment. Th2 cytokines decreased and increased the mRNA expression of SPT and DEGS2, respectively. Co-treatment with tofacitinib was able to counteract the Th2-mediated effects (Figure 2F). The mRNA levels of carbonic anhydrase 2 (CA2) and the transcription factor Peroxisome Proliferator-Activated Receptor-Gamma (PPARγ), genes involved in lipid metabolism [31], were up- and downregulated, respectively, after Th2 exposure. Tofacitinib counteracted the effects induced by the Th2 cytokines (Figure 2F).

### 3.4. Alteration of Lipid Profiles Induced by Th2 Cytokines and Modulating Effects of Tofacitinib

To understand the impact of Th2 cytokine-mediated signaling and the significance of JAK/STAT pathway inhibition in the 3D epidermal models, lipid extracts were analyzed using a combination of analytical approaches. The abundance profiles of FFAs, FOHs, and cholesterol in the HEEs were determined by GCMS. Polar and neutral lipid profiles were determined by LCMS. One-way ANOVA was performed on the lipidomics data to compare effects between groups. The test retrieved 51 lipid species that showed statistically significant differences among the 300 target species. Appendix A shows the results of one-way ANOVA with Tukey HSD post-hoc multiple comparisons. Normalized abundance (pmol/mg protein) was compared to vehicle and expressed as the average of six samples per group. To explore relationships and correlations between lipids, their expression profiles were organized in a hierarchical clustering (Figure 3A), showing trends in lipid modulation characteristics of each treatment. The dendrogram indicates linkage among regulated species. Three principal groups were identified based on the similarities in the changes of lipids’ abundance.

The first cluster is characterized by an overall lowering effect of Th2 cytokines, which was unmodified upon JAK/STAT inhibition. Lower abundance of hexosylceramides (HexCers), ultra-long FFAs, and saturated DGs was observed in all treatments. Interestingly, the reduction in HexCers abundance occurred to a statistically significant extent with tofacitinib alone (Appendix A). This finding may indicate an effect of tofacitinib on the recycling pathways of ceramides [32]. The Th2 cytokines significantly downregulated long-chain SFAs (Appendix A). The effects of tofacitinib were not remarkable (Figure 3B). The abundance of the detected DGs was significantly lower than the control regardless of the treatment (Appendix A).

The second cluster of the heatmap is defined by an induction stimulated by Th2 cytokines. It consists of LPC 20:4, sterol-like 458, some DGs and SMs, and PCs. Its overall abundance is lower in HEEs treated with tofacitinib compared to vehicle. Sterol-like species, whose identification awaits characterization, were observed together with their precursor cholesterol, whose concentration was only slightly affected, although tending to be lower in Th2 HEEs (Appendix A). Phospholipids account for the major components of the plasma membrane. Phosphocholine-containing lipids depicted in the heat map, i.e., PCs, LPCs, and SMs, are crucial elements of the plasma membrane and precursors in lipid synthesis [33] (Appendix A). Several members of the PC family were significantly modified in HEEs treated with Th2 cytokines (Figure 3C). Tofacitinib per se exerted no effect on the PC profile of HEEs. Irrespective of the direction of the modulation by Th2 cytokines, these effects were abrogated by tofacitinib. Interestingly, the PCs that were decreased and increased by Th2-type cytokines were dominated by the MUFAs FA 16:1 and FA 18:1 side chain (Appendix A). Other phospholipids, i.e., PEs, ether PEs, PGs, and PIs, were minimally affected by the treatments (Appendix A).

The third cluster of the heatmap represents all the species that were significantly downregulated by Th2 cytokines and that the inhibition of the JAK/STAT pathway by tofacitinib was able to revert the lipids-lowering effects of Th2 cytokines. Palmitoleic acid (FA 16:1n-7), a MUFA with 16 carbon atoms and a double bond (DB) at the delta-9 position, and FOH 17:1 were significantly reduced in the Th2-treated HEEs (Appendix A). The panel of metabolites determined by GCMS included several species that showed decreased concentration (pmol/mg protein) in all the treatments, although significance was not reached (Appendix A). As shown in Figure 3D, tofacitinib abrogated the Th2 effects on FA 16:1n-7. The observation of a lower abundance of FA 16:1-containing PCs could be related to the significantly reduced concentration of FA 16:1n-7 in its free form.

The amounts of two other sterol metabolites, tentatively assigned as lanosterol (sterol-like 498) and dihydrolanosterol (sterol-like 500) based on their EI-MS fragmentation spectrum, showed a similar trend (Appendix A).

The most significant effects on TGs are reported in Figure 3D and Appendix A. Interpretation of the MS/MS fragmentation spectra supported the prevalence of FA 16:1 as the bound FA in one or two sn positions of the TGs depleted by Th2 cytokines contained FA 16:1 (Appendix A) [33].

The observation of significantly depleted TGs suggested a disruption of lipid storage. Lipid droplets (LDs) are ubiquitous subcellular structures that serve the sequestration of TGs. We determined the mRNA levels of PLIN1 and PLIN2, proteins involved in the formation and regulation of intracellular LDs [34,35]. A significant reduction in the expression of both PLIN1 and PLIN2 was observed in Th2-treated HEEs, supporting the hypothesis of a Th2-driven depletion of the TG storage. Co-treatment with tofacitinib reversed the reduction in PLIN1 and PLIN2 mRNA expression caused by Th2 cytokines alone (Figure 3E). Since data in the literature show that FA oxidation is promoted in AD models, we preliminarily explored the hypothesis that in our model, FAs arising from TGs could undergo β-oxidation in mitochondria and peroxisomes [36,37]. To this end, we analyzed the mRNA expression of CPT1A, ACAT1, ACADS, genes encoding proteins involved in mitochondrial FA oxidation, and ACOX1 gene associated with peroxisomal FA oxidation [10,37,38,39,40]. Th2 cytokines significantly induced CPT1A and ACAT1 mRNA levels; co-treatment with tofacitinib abolished this effect (Figure 3F). ACADS and ACOX1 expression showed the same trend without reaching statistical significance. These results support the hypothesis of a link between the increased mitochondrial function observed in AD epidermis [37,41] and the increased utilization of FAs as energetic substrates under Th2 signaling.

Changes in concentrations of cholesterol sulfate and ceramides, although apparent, did not reach significance (Appendix A). The interpretation of the profiles of ceramide abundance in HEEs treated with tofacitinib and Th2 cytokines and their combination was not obvious. As expected, there was a trend towards decreased levels of ceramides in HEEs treated with Th2 cytokines (Appendix A). Nevertheless, CerNP species with 38 to 44 carbon atoms increased after Th2 cytokine treatment, as expected due to the significant upregulation of DEGS2. Ceramides bearing linoleic acid ester bound to omega hydroxylated long-chain FAs have been characterized in the epidermis (EOS, EODS, EOP, EOH) [42]. Several acylceramides showed a decrease in Th2-HEEs, which recovered with tofacitinib treatment, but significance was not reached. The changes in the concentration of two Cer[EOH] species, which decreased upon Th2 and recovered in presence of tofacitinib approached significance (Appendix A).

Since we expected to observe more pronounced modulation of ceramide profiles, also supported by the downmodulation of SPT mRNA, we undertook the analysis of ceramides according to their sphingoid base (SB). First, we checked the abundance of ceramides containing the sphingoid bases (SBs) sphingosine (S) and phytosphingosine (P) with chain lengths between 16 and 24 and between 16 and 20 carbon atoms, respectively, bound to a non-hydroxylated fatty acid (N). The results regarding the abundance of the main SBs were consistent with the overall profiles of the ceramides (Appendix A).

## 4. Discussion

Skin barrier defects in AD result from the dysregulation of multiple pathways. The incretion of Th2 cytokines in AD is at the interface between immune activation and EPB disruption. Lipids play a pivotal role in the water-holding capacity of the epidermis, and they are key components of the skin barrier, a complex system which encompasses physical, chemical, immunological, and microbiological aspects [43]. In AD, the hydrophobic barrier is not properly formed, due to mechanisms that have been only partly elucidated.

In this study, we report the effects of Th2 cytokines on lipid distribution using 3D human epidermal equivalents. We first assessed whether our 3D epidermal model mimicked the changes caused by stimulation with the cytokines IL-4 and IL-13. To this end, we performed morphometric analysis and evaluated specific differentiation markers relevant to the disease phenotype. As expected, FLG expression was reduced in HEEs treated with Th2 cytokines. In addition, we demonstrated a decrease in both pro-FLG and FLG localized in the SG and SC, respectively. However, we did not observe downregulation of IVL and LOR, contrary to what has been shown in other studies [8]. These discrepancies may be due either to differences in both the experimental models and the doses of cytokines used. The most innovative aspect of this study is the in-depth analysis of the lipid profile after stimulation with the cytokines IL-4 and IL-13 using lipidomic strategies. Despite some limitations due to differences in lipid composition and organization when compared to native skin, e.g., overall shorter chain length of FFAs, the 3D epidermal equivalent is a suitable model to study barrier properties in pathophysiological conditions reproduced in vitro [43,44,45]. Activation of the JAK/STAT axis by phosphorylation drives alterations in keratinocyte differentiation and abnormalities in skin lipids.

Ceramides, FFAs, and cholesterol and its conjugates (i.e., cholesterol sulfate and cholesterol esters) are key components of the EPB in the SC, which is the outmost epidermal layer. These three major lipid classes are present in equimolar abundance. Alterations in their molar ratio have been implicated in several skin diseases, particularly AD [46]. The hallmarks of lipid abnormalities in the SC in AD [47,48,49,50] include a shortening of the chain length of FAs bound in ceramides and a decrease in the absolute and relative abundance of acylceramides [51]. IL-4 markedly reduces the levels of long-chain ceramides in the epidermis by downregulating the expression of serine-palmitoyl transferase-2 (SPT2), acid sphingomyelinase (aSMase), and β-glucocerebrosidase (GCase) [52]. Ceramides play a crucial role in maintaining the homeostasis of the EPB and are also involved in cell signaling, proliferation, differentiation, and apoptosis in the human epidermis [46]. Ceramides are generated de novo in the endoplasmic reticulum by the condensation of serine and palmitoyl-CoA catalyzed by SPT [53,54].

Our results showed that Th2 cytokines decreased the mRNA expression of SPT. Although there was a trend towards the reduction of SPT end products, the effects were not straightforward. The experimental conditions used in this study, in particular the air–liquid phase, favored the detection of lipid changes that preceded the apparent change in ceramide abundance. Nevertheless, tofacitinib-mediated inhibition of the JAK/STAT pathway partially restores the modulation of both the SPT gene and ceramides caused by cytokines, suggesting a potential benefit of JAK/STAT blockade in reversing lipid perturbations induced by Th2 cytokines.

During keratinocyte differentiation, the total quantity of ceramides substantially increases [46,55]. To prevent intracellular ceramides from reaching cytotoxic levels, they are further processed into glucosylceramides and SMs and then transported into the extracellular space [46]. The decrease observed in the ceramide levels consequent to the Th2 cytokines is even more pronounced in the HexCer subgroup. HexCers include glucosylceramides, which are important components of the lamellar bodies, whose deficiency may contribute to the disturbed homeostasis of the EPB. It is likely that the JAK/STAT3 pathway is implicated in the biotransformation of ceramides via ceramide-glucosyltransferase, a key enzyme in glycosphingolipid synthesis [56]. SMs were minimally altered. Sphingomyelin synthase 2 (SGMS2) produces DGs and SMs by transferring phosphocholine from PC to the ceramide terminal OH group [57]. We found an increase and a decrease in the intermediate products PCs and DGs, respectively. This may be an indication of a disturbance in the SGMS2 pathway, which deserves further investigation in future studies.

FAs play a crucial role in the formation and dynamics of biological membranes, and they are essential for cell metabolism and energy balance [58]. In our study, the absolute quantities of FFAs, including MUFAs, were profiled. Conflicting results have been reported in the literature regarding the abundance of FFAs in AD. Both increased abundance of MUFAs and decreased levels of FA 16:1 and FA 18:1 have been described, together with increased susceptibility to S. aureus infection and dysfunction of the EPB [59,60]. Furthermore, MUFA depletion correlates with skin dryness in AD. Both natural moisturizing factors (NMFs) and skin surface lipids contribute to skin hydration. Recently, we described a significant depletion of MUFAs in the skin surface lipids derived from both sebum and SC [61]. Although palmitoleic acid (FA 16:1n-7) decreased significantly following Th2 cytokines, the involvement of the SCD1 desaturase pathway in the observed depletion of FA 16:1n-7 was unclear due to unchanged SCD1 mRNA levels. However, the possibility cannot be excluded that FA 16:1n-7 is also degraded by beta-oxidation [62]. Due to the importance of SCD1 in skin integrity, further investigation is needed to clarify the mechanisms causing FA 16:1n-7 depletion upon Th2 signaling [63]. The simultaneous addition of tofacitinib abrogated the delipidizing effects of Th2 cytokines. Indeed, we observed a significant recovery of FA 16:1n-7 when Th2 cytokines were co-administered with tofacitinib.

ELOVLs are responsible for elongating the FA chain by adding 2-carbon units. The ELOVL1 enzyme catalyzes the elongation of FA 18:0 to FA 26:0 and FA 18:1 to FA 22:1 when activated by coenzyme A (CoA) binding. The ELOVL3 and ELOVL4 enzymes are involved in the elongation of saturated FA 16:0 to FA 22:0-CoA and ultra-long-chain FAs (C26-36), respectively [64]. The observation of decreased expression of ELOVLs in 3D-HEEs treated with IL-4 and IL-13 is consistent with findings of their reduced expression in human SC from AD lesion areas [65,66]. Recent studies have identified significant metabolic changes in the SC and plasma of AD patients following treatment with dupilumab, a monoclonal antibody targeting the receptors for IL-4 and IL-13 [67,68]. Halting IL-4/IL-13 signaling was shown to revert the abundance of short chain NS-ceramides to normal and to restore the relative and absolute abundance of EOS-ceramides. The observed improvement following dupilumab treatment supports the involvement of ELOVL pathways in the downstream effects of IL-4/IL-13 signaling [68]. Inhibition of STAT phosphorylation by tofacitinib treatment restored the expression of ELOVL1, 3, and 4. The relationship between the levels of STAT6 and ELOVL3 and ELOVL6 has been reported [65]. Consistent with previous evidence, IL-4 and IL-13 promote the phosphorylation of STAT6 in human keratinocytes [28,69]. Inhibition of the JAK/STAT pathway by tofacitinib re-equilibrated the unbalanced lipid composition in human keratinocytes. However, the reported data are conflicting, and the mechanisms underlying the lipid chain shortening process remain unclear. The conflicting data on the expression and regulation of ELOVLs are likely due to the different experimental approaches used (e.g., in vitro models, cytokine doses, and treatment duration), each presenting limitations in mimicking the complexity of the AD skin microenvironment [66]. 

TG depletion is an initiating event and aggravating condition in AD [65,70]. The observation of decreased TGs is associated with the suppression of key genes involved in lipid synthesis, such as *DGAT1*, *DGAT2*, *FADS1*, and *ELOVL1* [71]. In our study, we observed that the TG species depleted upon exposure to Th2 cytokines through a JAK/STAT-dependent mechanism presented specific features, as demonstrated by the MS/MS data. A large body of evidence supports the role of JAK/STAT axis in the regulation of metabolic processes underlying energy expenditure and the turnover of lipid stores [72]. Cultured human keratinocytes vary their lipid composition in a density-dependent manner. Lipid neosynthesis is active before keratinocytes reach confluence, resulting in the accumulation of TGs in post-confluent cultures [73]. Normally, TGs are stored in LDs [74]. However, the presence of LDs in keratinocytes has only been studied in the context of epidermal dysfunction. PLINs bind to the surface of LDs and have both structural and regulatory functions [75]. The role of PLINs in the epidermis has been little investigated. Although the modulation of *PLIN1* and *PLIN2* transcripts in the Th2-treated HEEs awaits clarification, it has been observed to follow the changes in the TGs levels in this study. The reservoir of TGs is plastic and provides a pool of fatty acyl residues for phospholipid biosynthesis [73,76]. Defects in the catabolism of TGs have been described in humans with ichthyosis-bearing mutations in the *ABHD5/CGI-58* gene. Mice with a deficiency of genes involved in TG metabolism develop dysfunction of the EPB. Defective TG metabolism results in severe disruptions in the formation of acylceramides, essential for the build-up of the cornified lipid envelope in the epidermis [77]. Linoleic acid is mainly derived from TGs and is specifically incorporated into acylceramides [46].

There is evidence for metabolic relationships between glycerolipids, i.e., TGs and DGs, and PCs in mammalian cells. Specifically, the Kennedy pathway implicates the activation of phosphocholine with cytidine triphosphate (CTP), which is then transferred to DGs to produce PCs. Two enzymes of the Lands cycle, LPCAT1 and LPCAT2, synthesize PCs directly at the surface of the LDs where TGs are stored [78]. Our model showed a marked increase and decrease in PC species and TGs, respectively, when HEEs were treated with Th2 cytokines, suggesting a potential interaction between these lipid domains. Our findings support previous studies demonstrating a significantly high percentage of MUFAs bound in PCs in AD [79]. The significant increase in PCs is in line with the evidence of PC and phospholipid accumulation, which is an indicator of atopic pathogenic mechanism [70]. Indeed, it has been reported that although phospholipids are entirely degraded physiologically, they persist in AD epidermis [79]. This highlights the significance of phospholipid metabolism at the SG–SC interface in the potential release of FFAs, which support EPB function. It also indicates how the inadequate metabolism of phospholipids and their resultant accumulation represent a hallmark of AD [70]. Therefore, reversing PCs accumulation represents a benefit of tofacitinib.

Cholesterol is fundamental to the vital function of mammalian cells. In cutaneous tissues, cholesterol is engaged in the barrier architecture and acts as a precursor for steroid synthesis. Cholesterol levels are finely regulated, as even slight changes can have dramatic effects. While the HEEs treated with Th2 cytokines compensated for the Th2 effects on cholesterol, upstream precursors were affected, suggesting that the cholesterol biosynthetic pathway is a target of the Th2 signaling that might be controlled by JAK/STAT inhibition. It is important that future studies address the role of the PPARγ transcription factor in the lipid response to Th2 cytokines. Indeed, effects of perturbed PPARγ transcriptional activity may contribute, in different directions, to the overall mechanisms of action of tofacitinib.

## 5. Conclusions

Our results highlight that inhibition of the JAK/STAT pathway effectively abrogates lipid perturbations induced by Th2 cytokines (Figure 4). These findings suggest that JAK/STAT inhibitors, especially if applied topically, can correct the epidermal barrier lipid abnormalities induced by Th2 cytokines and be of great benefit to AD management.

## Figures and Tables

**Figure 1 cells-13-00760-f001:**
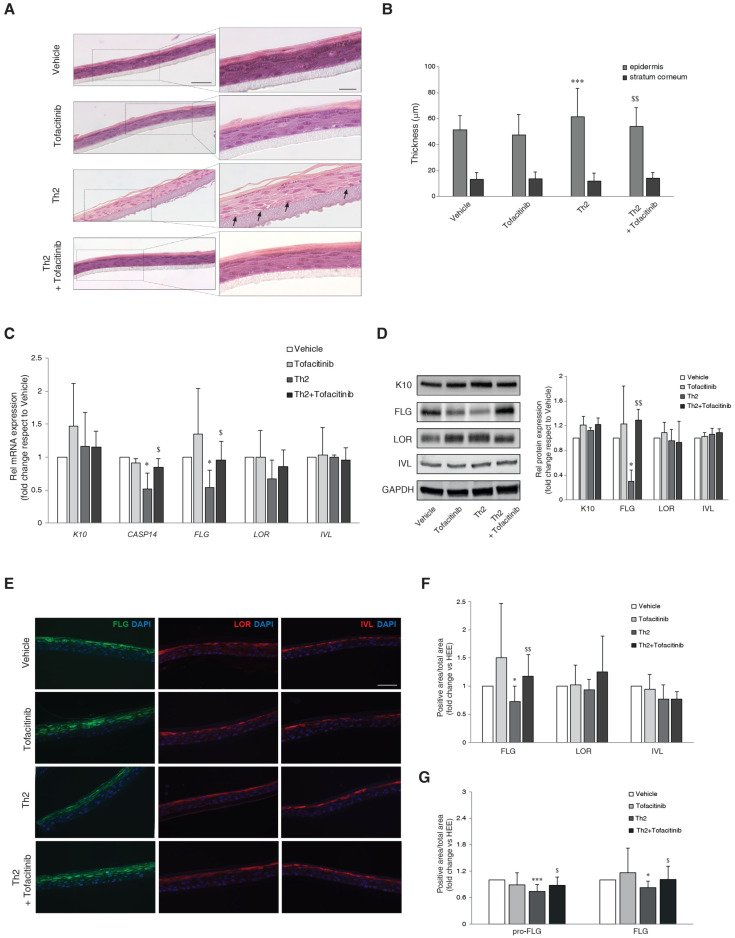
Effects of JAK/STAT inhibition by tofacitinib on the Th2-mediated changes in epidermal morphology and barrier protein/enzyme expression. (**A**) Hematoxylin and eosin staining (H&E) of paraffin-embedded 3D HEEs treated with vehicle Th2 (IL-4 and IL-13), and tofacitinib. Histological analysis of Th2-treated HEEs showed characteristic AD morphologic features such as epidermal thickening and increased spaces between adjacent keratinocytes (arrows). Scale bars: 50 µm, 20 µm. (**B**) Quantitative analysis of epidermal and SC thickness; *** *p* ˂ 0.001 vs. vehicle; ^$$^ *p* ˂ 0.01 vs. Th2. (**C**) Quantitative RT-PCR analysis of K10, CASP14, FLG, LOR, and IVL, performed on 3D HEEs and HEEs treated with Th2 cytokines and tofacitinib. All values of mRNA expression were normalized against the expression of GAPDH and were expressed relative to vehicle (taken as 1). Data represented the mean ± SD of three independent experiments; * *p* ˂ 0.05 vs. vehicle; ^$^ *p* ˂ 0.05 vs. Th2. (**D**) Western blot analysis of K10, FLG, LOR, and IVL protein expression performed on 3D HEEs and Th2-HEEs treated with tofacitinib. Representative blots are shown. GAPDH was used as endogenous loading control. Densitometric scanning of band intensities was performed to quantify the change in protein expression. Results were expressed as the fold change respect to vehicle (taken as 1-fold). Data represented the mean ± SD of three independent experiments; * *p* ˂ 0.05 vs. vehicle; ^$$^ *p* ˂ 0.01 vs. Th2. (**E**) Immunofluorescence and (**F**) quantitative analyses of FLG (green), LOR (red), and IVL (red); * *p* ˂ 0.05 vs. vehicle; ^$$^ *p* ˂ 0.01 vs. Th2, and of (**G**) pro-FLG and FLG on stratum granulosum and stratum corneum, respectively on serial sections of 3D HEEs and HEEs treated with tofacitinib and Th2 cytokines; * *p* ˂ 0.05 and *** *p* ˂ 0.001 vs. vehicle; ^$^ *p ˂* 0.05 vs. Th2. Nuclei were counterstained with DAPI (blue). Scale bar: 50 µm.

**Figure 2 cells-13-00760-f002:**
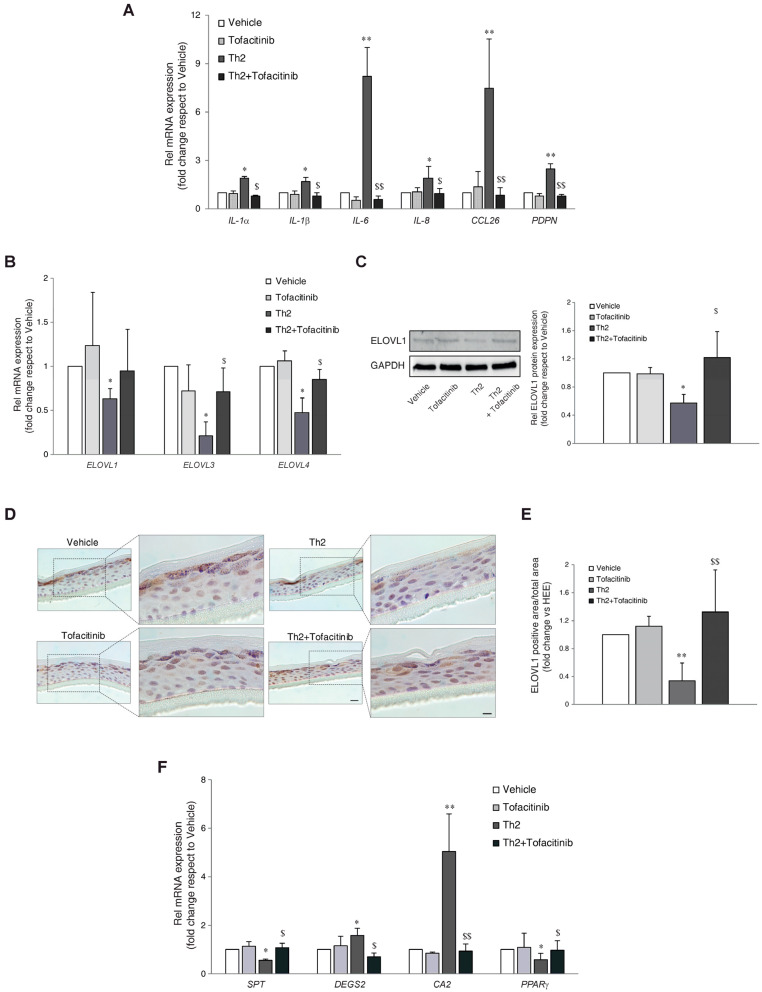
Effects of JAK/STAT inhibition by tofacitinib on the Th2-mediated changes in the expression of inflammatory and lipid genes. Quantitative RT-PCR analysis of (**A**) *IL-1α*, *IL-1β*, *IL-6*, *IL-8*, *CCL26*, and *PDPN* (**B**) *ELOVL1*, *ELOVL3*, and *ELOVL4*, performed on 3D HEEs and Th2-HEEs treated with tofacitinib. All values of mRNA expression were normalized against the expression of *GAPDH* and were expressed relative to vehicle (taken as 1). Data represented the mean ± SD of three independent experiments; * *p* ˂ 0.05, ** *p* ˂ 0.01 vs. vehicle; ^$^ *p* ˂0.05, ^$$^ *p* ˂ 0.01 vs. Th2. (**C**) Western blot analysis of ELOV1 protein expression performed on 3D HEEs and Th2-HEEs treated with tofacitinib. Representative blots are shown. GAPDH was used as the loading control. Densitometric scanning of band intensities was performed to quantify the change in protein expression. Results were expressed as the fold change respect to vehicle (taken as 1-fold). Data represented the mean ± SD of three independent experiments; * *p* ˂ 0.05 vs. vehicle; ^$^ *p* ˂ 0.05 vs. Th2; (**D**) Immunohistochemical and (**E**) quantitative analyses of ELOVL1 on 3D HEEs and Th2-HEEs treated with tofacitinib. Nuclei were counterstained with hematoxylin. Scale bars: 20 µm, 10 µm. ** *p* ˂ 0.01 vs. vehicle; ^$$^ *p* ˂0.01 vs. Th2. (**F**) Quantitative RT-PCR analysis of *SPT*, *DEGS2*, *CA2*, and *PPARγ*, performed on 3D HEEs and Th2-HEEs treated with tofacitinib. All values of mRNA expression were normalized against the expression of *GAPDH* and were expressed relative to vehicle (taken as 1). Data represented the mean ± SD of three independent experiments; * *p* ˂ 0.05, ** *p* ˂ 0.01 vs. vehicle; ^$^ *p* ˂ 0.05, ^$$^ *p* ˂ 0.01 vs. Th2.

**Figure 3 cells-13-00760-f003:**
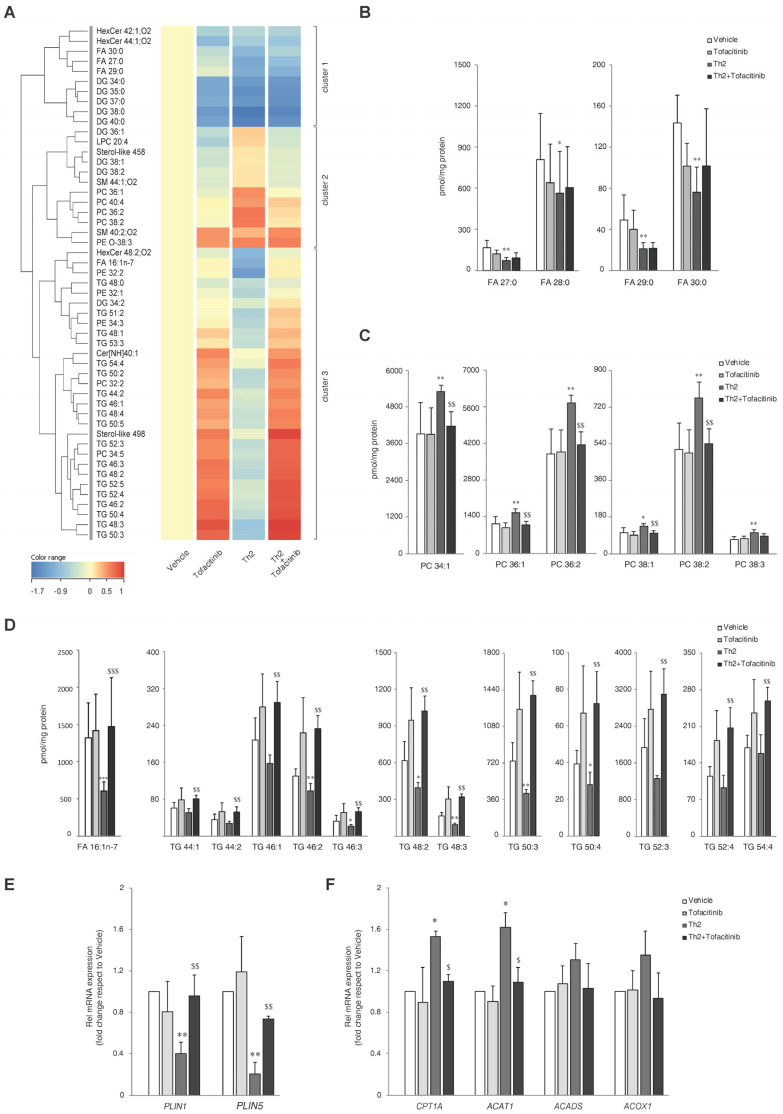
Th2 cytokines IL-4 and IL-13 induce lipid changes through the activation of JAK/STAT pathway. (**A**) Hierarchical clustering of 51 lipid species resulted significantly modulated in the one-way ANOVA applied to the amounts of the 300 characterized lipids. (**B**) Ultra-long chain SFAs determined by LCMS. Profiles of abundance of (**C**) PCs determined by HILIC-MS and of (**D**) Palmitoleic acid (FA 16:1n-7) determined by GCMS and TGs determined by RP-LCMS in lipid extracts of 3D HEEs treated with vehicle, tofacitinib, Th2 cytokines, and combined tofacitinib and Th2 cytokines. Molar amounts of individual lipids were calculated against same-class deuterated internal standards and were normalized by the protein concentration and expressed as pmol/mg protein. Real Time RT-PCR analysis of (**E**) *PLIN1* and *PLIN2*, (**F**) *CPT1α*, *ACAT1*, *ACADS*, and *ACOX1*, performed on 3D HEEs and HEEs treated with Th2 cytokines and tofacitinib. All values of mRNA expression were normalized against the expression of *GAPDH* and were expressed as relative to the vehicle (taken as 1). Data represented the mean ± SD of three independent experiments; * *p* ˂ 0.05, ** *p* ˂ 0.01 vs. vehicle; ^$^ *p* ˂ 0.05, ^$$^ *p* ˂ 0.01 vs. Th2.

**Figure 4 cells-13-00760-f004:**
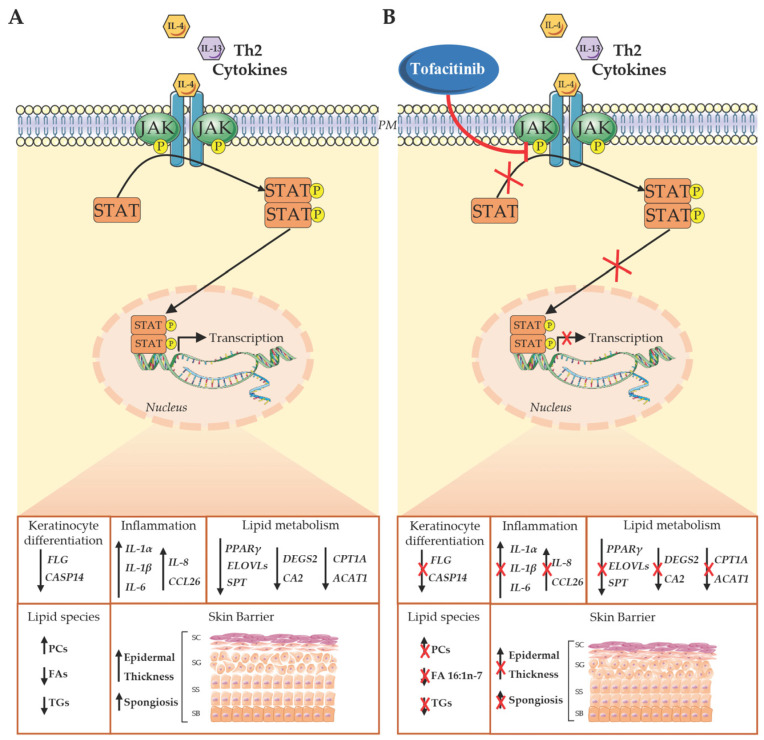
Effect of Th2 cytokines and tofacitinib on JAK/STAT signaling. (**A**) After stimulation with Th2 cytokines (IL-4 and IL-13), activated JAK phosphorylates STATs, which dimerize and translocate to the nucleus. Upon binding to DNA, STATs regulate the transcription of selected genes involved in keratinocyte differentiation, inflammation, and lipid metabolism. As a result, lipid profiles are altered, and the epidermis becomes thicker due to an increase of the space between adjacent keratinocytes. (**B**) Tofacitinib suppresses STAT phosphorylation through JAK inhibition, counteracting Th2-dependent alterations. PM: plasma membrane; SC: stratum corneum; SG: stratum granulosum; SS: stratum spinosum; SB: stratum basale.

**Table 1 cells-13-00760-t001:** Pathway of JAK-STAT in AD. Adapted from Amano et al. [9].

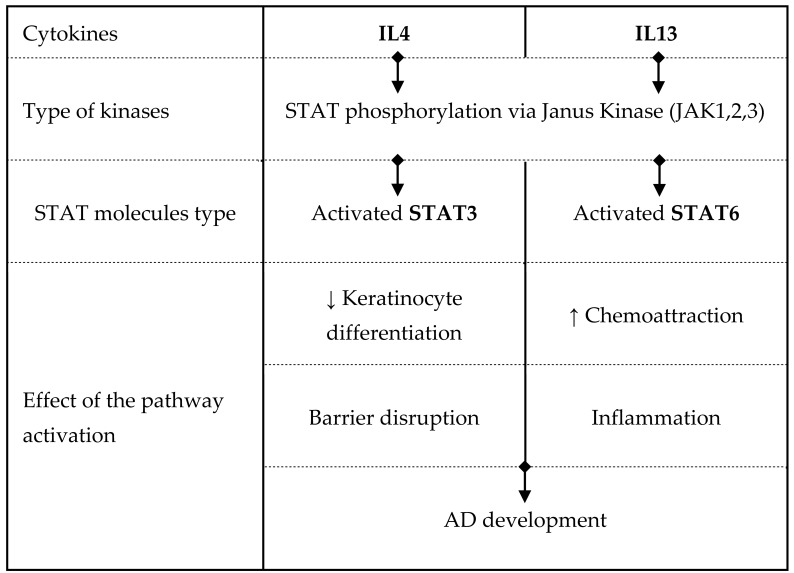

## Data Availability

All the data described are contained within the manuscript.

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
