# Peer review of "JAK/STAT Inhibition Normalizes Lipid Composition in 3D Human Epidermal Equivalents Challenged with Th2 Cytokines"

_cells, 2024, doi:10.3390/cells13090760_

Round 1

Reviewer 1 Report

Comments and Suggestions for Authors

The manuscript is an useful contribution to the journal. However, a few major points should be addressed in order to increase the value of the manuscript, as follows:

--An image depicting JAK/STAT activation pathways, including dimerization  and phosphorylation events that fundament the activation steps, as well as the exact places of inhibiton exercited by the JAK/STAT inhibitors could increase the value of the manuscript and would benefit the reader.   --Also, the manuscript should mention the various pathways inhibited by JAK inhibitors, to present the wider context in which the current research is placed (e.g inserting a table linking the particular cytokines ---> type of kinases---> STATs molecules type--->already known effect of the inhibition of the pathway.   --Moreover, I consider it would be worth mentioning the FDA approved inhibitors for either systemic or topical use.   --In this context, the authors should explain clearly their choice of the JAK inhibitor employed in the current study, as compared to other JAK inhibitors available.     Comments on the Quality of English Language

Minor editing of English required.

Author Response

Response to Reviewer 1

We are thankful to this Reviewer for the insightful suggestions. The raised comments have been categorized in three parts, which are addressed correspondingly.

The references added to the bibliography are written in italic/orange.

(1) As suggested, an image depicting the JAK/STAT activation pathway and the sites in the pathway targeted by tofacitinib has been added.

(2) As suggested, mention of the pathways inhibited by the JAK/STAT inhibitors has been added based on the recent literature:

Nakashima C, Yanagihara S, Otsuka A. Innovation in the treatment of atopic dermatitis: Emerging topical and oral Janus kinase inhibitors. Allergol Int. 2022 Jan;71(1):40-46. doi: 10.1016/j.alit.2021.10.004. Epub 2021 Nov 21. PMID: 34815171.

 The table reported in Figure 1 has been generated (with permission) starting from the Figure 7 in:

 Amano W, Nakajima S, Kunugi H, Numata Y, Kitoh A, Egawa G, Dainichi T, Honda T, Otsuka A, Kimoto Y, Yamamoto Y, Tanimoto A, Matsushita M, Miyachi Y, Kabashima K. The Janus kinase inhibitor JTE-052 improves skin barrier function through suppressing signal transducer and activator of transcription 3 signaling. J Allergy Clin Immunol. 2015 Sep;136(3):667-677.e7. doi: 10.1016/j.jaci.2015.03.051. Epub 2015 Jun 24. PMID: 26115905.

(3) As mentioned in the added citation: ‘Tofacitinib is a first-generation small molecules designed to selectively block JAK1 and JAK3, and to a lesser extent, JAK2 and TYK2.’ Tofacitinib has been indicated for topical treatment for mild-moderate AD. The status of approval by the US Food and Drug administration (FDA) of JAK inhibitors is reviewed in:

Huang IH, Chung WH, Wu PC, Chen CB. JAK-STAT signaling pathway in the pathogenesis of atopic dermatitis: An updated review. Front Immunol. 2022 Dec 8;13:1068260. doi: 10.3389/fimmu.2022.1068260. PMID: 36569854; PMCID: PMC9773077.

At the best of our knowledge this is the most recent review on the FDA-approved JAK inhibitors. Thus, the reference above has been added to the manuscript.

The choice of Tofacitinib to investigate effects of JAK/STAT inhibition on lipid perturbation has different fundaments: Tofacitinib is used for the topical treatment of AD, due to its effects exerted directly on the epidermal cells. Effects of tofacitinib on the epidermal components other than lipids, has been characterized in 3D epidermal models. At the time of the project design, tofacitinib was commercially available from a supplier of chemicals.

We learn from this comment that characterization of individual JAK inhibitors for their effects of keratinocytes lipidome may have important implications in the definition of clinical outcomes. The 3D epidermal equivalents may offer suitable tools to investigate in vitro the potential of JAK inhibitors candidate for the approval in AD, to be tested for their effects on the epidermal barrier perturbed by Th2 cytokines.

Reviewer 2 Report

Comments and Suggestions for Authors

The author reported that JAK/STAT inhibition normalizes lipid composition in 3D human epidermal equivalents challenged with Th2 cytokines. However, the major concern is whether the author truly induced an in vitro AD-like 3D epidermal equivalents model. The H&E staining result and statistical chart in Figure 1 are entirely unconvincing to the reviewer. Additionally, the lipid analysis study lacks information on the lipid extraction rate, standard curve, limit of detection, and limit of quantification for each selected lipid.

Author Response

Response to Reviewer 2

(1) We thank this reviewer for highlighting the limited fidelity of in vitro models in reproducing disease. The main limitations are due to the difficulties in incorporating the different dermal/epidermal cell types that create the complex microenvironment enriched in humoral factors responsible for the AD phenotype. However, 3D models are particularly useful for studying selected factors for their intrinsic effects, excluding possible interactions that occur in more complex systems. Several studies use the human epidermal equivalents to obtain keratinocyte full differentiation and to reproduce features of AD skin by IL-4 and IL-13 stimulation. To respond to the reviewer’s suggestion, we have replaced Figure 1A with a new image of Th2-treated HEEs to better show the presence of spongiosis and increased thickness of the epidermal layers. Measurements using the Zeiss quantification program showed a slight increase in epidermal thickness. However, as reported, this was statistically significant. Although AD pathomechanisms are the obvious link to the effects of Th2-type cytokines, the experimental design allowed the observation of effects on lipid modifications in the epidermal models that have not been previously reported.

(2) Accurate quantification based on calibration curves of authentic standards is feasible only for the targeted analysis of a few analytes. Although the number of authentic standards available is expanding, prediction and purchase of the detected species is time and cost ineffective. In the lipidomic suspect screening, concepts arising from absolute quantification, such as recovery, LOD, LOQ, are traded with large scale semi-quantitative analysis. In line with the current indications, and our previous study in 3D organotypics (references below added in the bibliography) our approach applied the principles of the semiquantitative semi-targeted analysis by incorporating a large range of internal standards covering the major cell lipids, as detailed in the supplementary Table S1 in the additional information. The presence of same class internal standard guaranteed accurate compensation in control and treated cells.

Züllig T, Trötzmüller M, Köfeler HC. Lipidomics from sample preparation to data analysis: a primer. Anal Bioanal Chem. 2020 Apr;412(10):2191-2209. doi: 10.1007/s00216-019-02241-y. Epub 2019 Dec 10. PMID: 31820027; PMCID: PMC7118050. 

Köfeler HC, Ahrends R, Baker ES, Ekroos K, Han X, Hoffmann N, Holčapek M, Wenk MR, Liebisch G. Recommendations for good practice in MS-based lipidomics. J Lipid Res. 2021;62:100138. doi: 10.1016/j.jlr.2021.100138. Epub 2021 Oct 16. PMID: 34662536; PMCID: PMC8585648.

McGeoghan F, Camera E, Maiellaro M, Menon M, Huang M, Dewan P, Ziaj S, Caley MP, Donaldson M, Enright AJ, O'Toole EA. RNA sequencing and lipidomics uncovers novel pathomechanisms in recessive X-linked ichthyosis. Front Mol Biosci. 2023 Jun 7;10:1176802. doi: 10.3389/fmolb.2023.1176802. PMID: 37363400; PMCID: PMC10285781.

Reviewer 3 Report

Comments and Suggestions for Authors

Dear Authors,

I appreciate the opportunity to review the study "JAK/STAT inhibition normalizes lipid composition in 3D human epidermal equivalents challenged with Th2 cytokines" by Flori et al., which investigated how Th2 cytokines affect the lipid composition in 3D human epidermal equivalents and found that inhibition of JAK/STAT can prevent this lipid disturbance.

This is a relevant topic to dermatology and immunology, as it investigates the impact of Th2 cytokines on lipid composition in 3D human epidermal equivalents, which is an area of interest given the role of these cytokines in atopic dermatitis (AD) and the lipid metabolism dysregulation observed in this condition.

The study's strength lies in exploring the potential of JAK/STAT inhibition, specifically with tofacitinib, in preventing lipid disturbances caused by Th2 signaling. Additionally, the study addresses a specific gap in the field by examining the molecular mechanisms underlying lipid alterations in AD and assessing targeted pharmacological interventions to restore lipid homeostasis.

Finally, the study contributes valuable insights into understanding and potentially treating lipid-related aspects of AD, which could lead to the development of novel therapeutic strategies.

Therefore, I recommend the publication of this study.

Author Response

Response to Reviewer 3

We are thankful for the positive feedback of this Reviewer, who provided comments in support of the relevance of the topic, and highlighted a knowledge gap addressed with the analysis of keratinocytes’ lipidome.

Round 2

Reviewer 1 Report

Comments and Suggestions for Authors

The manuscript has been revised according to the suggestion. It has an improved version and could br publushed in current form. 

Comments on the Quality of English Language

Minor editing required

Author Response

Response to Reviewer 1

The paper has been carefully re-read for the English editing. Following text editing and duplication minimization, the changed parts are highlighted in blue.

Reviewer 2 Report

Comments and Suggestions for Authors

The author should rewrite and reduce the rate of plagiarism.

Author Response

Response to Reviewer 2

Thank you for supporting us with the details on the duplication rate found in the manuscript. We have rewritten several parts of the manuscript. Where possible, we changed the text of the M&M, although the source of duplication in the M&M was partly linked to our own articles and to the submitted preprint of the same manuscript. All changes made to the manuscript were highlighted in blue.